# The Use of Salivary Levels of Matrix Metalloproteinases as an Adjuvant Method in the Early Diagnosis of Oral Squamous Cell Carcinoma: A Narrative Literature Review

**Monica Monea** [1,*] **and Anca Maria Pop** [2]

1 Department of Odontology and Oral Pathology, George Emil Palade University of Medicine, Pharmacy, Science, and Technology of Târgu Mureș, 540139 Târgu Mureș, Romania

2 Faculty of Medicine, George Emil Palade University of Medicine, Pharmacy, Science, and Technology of Târgu Mureș, 540139 Târgu Mureș, Romania

\* Correspondence: monica.monea@umfst.ro

**Abstract:** Oral squamous cell carcinoma (OSCC) is an aggressive malignancy with increased mortality, in which the early diagnosis is the most important step in increasing patients' survival rate. Extensive research has evaluated the role of saliva as a source of diagnostic biomarkers, among which matrix metalloproteinases (MMPs) have shown a valuable potential for detecting even early stages of OSCC. The aim of this review was to present recent clinical data regarding the significance of salivary MMPs in the detection of early malignant transformation of the oral mucosa. A narrative review was conducted on articles published in PubMed, Cochrane Library, Web of Science, EBSCO and SciELO databases, using specific terms. Our search revealed that MMP-1, MMP-2, MMP-3, MMP-8, MMP-9, MMP-10, MMP-12 and MMP-13 had significantly higher levels in saliva from patients with OSCC compared to controls. However, the strength of evidence is limited, as most information regarding their use as adjuvant diagnostic tools for OSCC comes from studies with a low number of participants, variable methodologies for saliva sampling and diagnostic assays, and insufficient adjustment for all covariates. MMP-1, MMP-3 and MMP-9 were considered the most promising candidates for salivary diagnosis of OSCC, but larger studies are needed in order to validate their clinical application.

**Keywords:** early diagnosis of cancer; oral squamous cell carcinoma; salivary proteins and peptides; matrix metalloproteinases



## 1. Introduction

Oral cancer is the sixth most frequent form of cancer worldwide, showing a higher prevalence in developing countries [1,2]. The most affected individuals are men over 50 years of age [3], with a low socioeconomic status [4] and exposed to risk factors such as smoking, chronic use of alcohol, poor oral hygiene, lack of proper dental treatments or human papilloma virus (HPV) infection [5]. Furthermore, periodontal disease was also documented to increase the risk of oral cancer, as the severity of periodontal inflammation was reported to be correlated with the development of oral squamous cell carcinoma (OSCC) [6–8].

Oral cancer is known to be highly invasive and one of the most debilitating forms of cancer, often leading to functional difficulties (speaking, swallowing) or disfiguration [9]. The morbidity and mortality have still remained high, despite considerable advances in diagnosis and treatment [10]. OSCC accounts for 90% of all oral cancers [11] and should be considered a distinct entity that must be separated from oropharynx cancers due to differences regarding the targeted population [12] and incidence disparities [13] alongside the development of specific detection and control strategies [14]. Based on data from the Global Cancer Observatory, the five-year prevalence of OSCC was almost 1 million; moreover, it should be taken into account that the actual burden of OSCC may be

underestimated due to deficient reporting of data from developing countries or improper documentation of lesions [15].

The incidence of OSCC continues to be high in low- and middle-income countries due to the increased popularity of homemade alcoholic drinks that contain high quantities of acetaldehyde, along with tobacco consumption in the form of cigarettes, but also smokeless tobacco products. In high-income countries, the incidence rates of OSCC did not decline significantly, mainly due to the increasing prevalence of HPV-associated OSCC [16]. Despite new technologies for diagnosis and treatment, the 5-year survival rate for OSCC remains under 40% for the cases diagnosed in stages III and IV, while the 5-year recurrence-free survival rates for cases identified in stages I and IV are 80% and 20%, respectively [17]. If a carcinoma is detected in the early localized stages (stage I, II), the prognosis is good; the patients can receive surgical treatment, with or without radiotherapy and chemotherapy, allowing complete remission and a 5-year survival rate of 85% [18,19]. However, in patients with regional and distant metastases, the 5-year survival rate drops to 42% and 17%, respectively [20]. Unfortunately, around 60% of cases are diagnosed in late stages (stage III and IV) when about 50% of patients have already developed metastases [21]. Due to the fact that these rates have not changed over the last decade, it was suggested that the best way to increase the survival rates of OSCC is to focus on its early detection [22].

OSCC is frequently preceded by potentially malignant oral lesions (PMOLs), which could be the target of early diagnosis [23]. The current gold standard for diagnosis is represented by biopsy followed by a histopathologic exam, which is an invasive, painful, and time-consuming technique. Moreover, in large PMOLs, several biopsies from multiple sites are sometimes needed in order to confirm or exclude malignant transformation. Therefore, other less invasive diagnostic tests have been proposed, which could also provide a more rapid result [24].

Due to its proximity to the tumor microenvironment, saliva was considered a promising candidate as a diagnostic fluid for OSCC, enabling the detection of cancer biomarkers in a simple, non-invasive, easy-to-handle and -store technique [25]. Among the vast panel of detectable molecules, matrix metalloproteinases (MMPs) were extensively linked to the development of various types of malignancies, including OSCC [26]. This review aimed to provide up-to-date information regarding the role of salivary levels of MMPs as complementary tools for the early diagnosis of PMOLs and OSCC, based on data available on these molecules' contribution to OSCC progression and on the use of saliva as a diagnostic fluid.

## 2. Data Collection

We conducted a narrative literature review in order to find a correlation between the salivary levels of MMPs and early OSCC diagnosis or progression of PMOLs to OSCC, based on articles listed in electronic databases such as PubMed, Cochrane Library, Web of Science, EBSCO and SciELO, using the following keywords: (early diagnosis of oral cancer OR potentially malignant oral lesions OR oral squamous cell carcinoma) AND (salivary levels of MMPs OR salivary proteins OR salivary biomarkers). As filters, we applied original articles (clinical trials, prospective and cross-sectional studies) published in the last ten years. The search was conducted between August and October 2022 by two independent reviewers who evaluated all identified articles. After the primary search, the relevance to the topic was assessed based on the title and abstract. The articles considered eligible were downloaded and extensively evaluated. Moreover, we also consulted the reference list of the selected papers in order to find further articles relevant to our topic. We excluded duplicates, and all disagreements regarding the included data were solved based on discussions between the evaluators.

## 3. Clinical Significance of Salivary Levels of MMPs in OSCC Detection

The key findings of the 11 articles included in our review are summarized in Table 1. These are all clinical studies published over the last decade in which salivary levels of

MMPs were measured in patients with OSCC compared to patients with PMOLs and healthy controls.

**Table 1.** Clinical findings of the reviewed studies.

| Authors | Evaluated Biomarkers | Study Participants | Laboratory Analysis | Results | Conclusion |
|---|---|---|---|---|---|
| Stott-Miller et al., 2011 [26] | MMP-1, MMP-3 | 60 primary OSCC cases, 15 cases of oral dysplastic lesions and 25 controls | ELISA | Higher salivary concentration of MMPs in OSCC cases compared to controls (6.2 times higher for MMP-1 and 14.8 times higher for MMP-3); Stronger results for both MMP-1 and MMP-3 in oral cavity cancer versus controls, rather than in oropharyngeal cancer versus controls; Salivary concentrations of MMP-1 and MMP-3 correlated with tumor stage. | MMPs may be very useful in monitoring dysplasia progression to OSCC; Salivary MMP-1 can be an important biomarker of OSCC development. |
| Chang et al., 2020 [27] | MMP-1 | 269 primary OSCC cases, 578 PMOLs and 313 healthy controls | ELISA | Higher salivary concentration of MMP-1 in OSCC cases than in non-cancerous patients (PMOLs and healthy controls); Satisfactory reliability of MMP-1 in distinguishing OSCC from non-tumor lesions; Good power of discrimination between OSCC located on the tongue, cheek mucosa, gum and multiple sites of the oral cavity from non-tumor lesions; Higher salivary concentrations of MMP-1 in high-risk PMOLs compared to low-risk PMOLs and healthy controls; Correlation of the salivary levels of MMP-1 with tumor size and lymph node metastasis. | Useful salivary biomarker in evaluating malignant transformation; Promising potential in early-stage screening of patients with increased risk for OSCC development; Important biomarker of poor prognosis. |
| Yu et al., 2016 [28] | MMP-1, MMP-3, MMP-9, annexin-2 (ANXA2), Heat Shock Protein Family A (Hsp70) Member 5 (HSPA5) and other salivary proteins | 131 OSCC cases, 233 PMOLs and 96 healthy controls | Liquid chromatography–multiple reaction monitoring-mass spectrometry | MMP-1 and kininogen 1 (KNG 1) had the highest salivary concentrations in OSCC patients among the analyzed salivary proteins; The sensitivity and specificity of the analyzed MMPs in differentiating OSCC from healthy controls and low-risk PMOLs were: 69.5% and 95% for MMP-1, 62.6% and 76.9% for MMP-3 and 75.6% and 60.3% for MMP-9; | An identified panel of 4 salivary biomarkers (MMP-1, KNG 1, ANXA2 and HSPA5) may represent a promising tool in the diagnosis of OSCC and in the monitoring of malignant transformation occurring in high-risk PMOLs. |

**Table 1.** *Cont.*

| Authors | Evaluated Biomarkers | Study Participants | Laboratory Analysis | Results | Conclusion |
|---------|---------------------|--------------------|--------------------|---------|------------|
| Feng et al., 2019 [29] | MMP-1, MMP-2, MMP-3, MMP-7, MMP-8, MMP-9, MMP-10, MMP-12, MMP-13 and other salivary proteins | 20 OSCC cases, 20 oral benign masses (OBM), 20 mild chronic periodontal disease (CPD) and 20 healthy controls | ELISA | MMP-1, MMP-2, MMP-3, MMP-10, MMP-12, MMP-13 were detected only in the saliva of patients with OSCC; MMP-1, MMP-2, MMP-10, MMP-12 along with cathepsin V, A disintegrin and a metalloprotease 9 (ADAM9) and kallikrein 5 has higher salivary concentration in OSCC patients compared to OBM, CPD and healthy controls. | By evaluating the combined sensitivity and specificity, the concomitant use of ADAM9, cathepsin V and kallikrein 5 was the most promising candidate for diagnosing OSCC. |
| Cai et al., 2022 [30] | MMP-1, MMP-2, MMP-3, MMP-8, MMP-9, MMP-10, MMP-13, Hepatocyte growth factor (HGF) and other salivary proteins | 8 OSCC cases and 8 healthy controls | Protein chip array | Higher salivary concentrations of MMP-1, MMP-3, MMP-8, MMP-9, MMP-10 and MMP-13 in OSCC patients compared to controls, the most significant increases being observed for MMP-1, MMP-3 and MMP-13; No difference between salivary concentration of MMP-2 in OSCC compared to controls; MMP-1, MMP-3 and MMP-13 were not detected in the saliva of healthy controls. The salivary levels of HGF and MMP-9 significantly differed between OSCC patients and healthy subjects. | Both HGF and MMP-9 can be useful biomarkers for the diagnosis and prognosis of OSCC. |
| Agha-Hosseini et al., 2015 [31] | MMP-13 | 20 OSCC cases and 30 oral lichen planus (OLP) | ELISA | No differences between salivary concentrations of MMP-13 in OSCC patients compared to OLP. | Salivary MMP-13 may not be a valuable adjuvant in the diagnosis and screening of OSCC. |
| Ghallab and Shaker, 2017 [32] | MMP-9 and chemerin | 15 early-stage OSCC cases, 15 PMOLs and 15 healthy controls | ELISA | Higher salivary levels of MMP-9 and chemerin in patients with OSCC compared to those with PMOLs and healthy subjects; Higher salivary levels of MMP-9 and chemerin in PMOLs compared to healthy subjects; Salivary MMP-9 had the greatest accuracy in distinguishing PMOLs from OSCC (sensitivity 100% and specificity 93%). | Salivary MMP-9 and chemerin can be used as adjuvant tools in the early diagnosis of OSCC and PMOLs. |
| Peisker et al., 2017 [33] | MMP-9 | 30 OSCC cases and 30 healthy controls | ELISA | Higher salivary levels of MMP-9 in OSCC patients compared to healthy controls. Salivary MMP-9 had 100% sensitivity and 26.7% specificity in diagnosing OSCC. | Salivary MMP-9 may be a complementary tool for the early diagnosis of OSCC. |

**Table 1.** *Cont.*

| Authors | Evaluated Biomarkers | Study Participants | Laboratory Analysis | Results | Conclusion |
|---------|---------------------|--------------------|--------------------|---------|-----------|
| Shin et al., 2021 [34] | MMP-9 and 8-hydroxydeoxy-guanosine (8-OHdG) | 106 OSCC cases and 212 healthy controls | ELISA | The salivary levels of MMP-9 in OSCC patients were 17 times higher than those of healthy controls; After surgical excision of the tumor, salivary MMP-9 decreased by 80% in the first nine months but was still higher in comparison with the value in healthy controls; The diagnostic ability of salivary MMP-9 after adjusting covariates showed 97.2% sensitivity and 94.2% specificity; | Salivary MMP-9 may be used in the early diagnosis and screening of OSCC. |
| Smriti et al., 2020 [35] | MMP-9 | 24 OSCC cases, 20 PMOLs, 22 subjects consuming tobacco and 22 healthy controls | ELISA | Higher levels of MMP-9 in the saliva of patients with OSCC and PMOLs compared to tobacco users and healthy controls; Salivary levels of MMP-9 increased according to tumor stage and were higher in poorly differentiated tumors; | Salivary MMP-9 can aid in the diagnosis of OSCC and PMOLs. |
| Saleem et al., 2021 [36] | MMP-12 | 30 OSCC cases, 30 patients with oral submucous fibrosis (OSF) and 30 healthy controls | ELISA | Higher levels of MMP-12 in the saliva of patients with OSCC and OSF compared to healthy controls and also in OSCC group compared to OSF group; Salivary MMP-12 had 100% sensitivity and 100% specificity in detecting OSF and OSCC. | Salivary MMP-12 can be an adjuvant method in the early diagnosis of OSF and OSCC. |

## 4. The Role of Saliva in Oral Cancer Detection

Human saliva contains peptides, proteins, electrolytes, and organic and inorganic compounds secreted by salivary glands alongside the fluids from the gingival sulcus and the transudates of the oral mucosa [11]. Regarded as "the mirror of the body" [37], it is highly informative, non-invasive, very accessible, safe to handle and easy to store fluid, which are qualities that make it an ideal diagnostic fluid [38,39]. The evaluation of biomarkers present in saliva was transferred from experimental to clinical practice as a result of improved genomic, metabolomic, proteomic and transcriptomic technologies [40,41], which allowed sensitive and specific detection of proteins and nucleic acid targets [38,42–45]. The current gold standard for OSCC diagnosis is based on clinical examination and histopathologic analysis, but cases located in hidden areas of the oral cavity might remain undetected [46,47]. Therefore, the use of sensitive and specific biomarkers could improve the detection of OSCC and allow better screening of high-risk patients [44]. Due to its high content in biomarkers, saliva was extensively studied during the last decade as a source for a "liquid biopsy" for the diagnosis and prognosis of OSCC [48].

As it shares direct contact with OSCC and PMOLs, saliva was the first examination choice in the screening of biomarkers, as it reflects the molecular environment of these lesions better than distant fluids such as blood [44]. Although saliva was considered a viable alternative or adjuvant diagnostic method, it should nevertheless be followed by biopsy confirmation [42]. Moreover, its beneficial role in the management of OSCC was also enhanced by its feasibility in predicting the post-therapy prognosis [43]. However, a limitation of the use of salivary biomarkers is related to their reduced concentrations

(100–1000 fold) compared to plasma, but this aspect is not particularly crucial in the detection of OSCC because these molecules are secreted locally in the proximity of the primary tumor [49]. On the other hand, the standardization of salivary tests requires high specificity and sensitivity of the selected molecules [50] in order to differentiate PMOLs and OSCC from other inflammatory conditions, such as gingivitis, periodontitis or infections that might affect the levels of salivary biomarkers [51]. The presence of inflammation could result in false-positive responses, reducing the diagnostic value of the biomarkers in PMOLs and OSCC. Therefore, further research is required in order to confirm the reliability of these tests based on salivary biomarkers and to validate them for future clinical applications.

## 5. The Key Role of MMPs in the Development of the Tumor Microenvironment of OSCC

MMPs are proteolytic enzymes, involved especially in the dissolution of the extracellular matrix (ECM) components. Recently, they have been acknowledged as biomarkers for various diseases, mainly cancerous pathologies [52]. Physiologically, MMPs participate in processes such as cellular differentiation and mobility, angiogenesis, apoptosis and tissue remodeling. However, the deregulation of their function promotes the development of several pathologies associated with tissue destruction, ECM loosening and fibrosis [53]. Studying MMPs in OSCC has gained interest during the last decade, as these were proven to have major implications in pathological conditions of the oral cavity with intense degradation of ECM, such as periodontal disease and other oral malignancies [54].

The key step mainly regulated through MMPs in the development of OSCC is the degradation of the ECM, which induces a crucial change in the cellular phenotype. This leads to epithelial–mesenchymal transition, a phenomenon characterized by loss of cell polarity and cell-to-cell adhesion, which facilitates the invasive potential of tumor cells [55]. Moreover, by degrading collagen, MMPs reveal normally hidden sites in the ECM, allowing integrins to interact with its components [56,57]. The digestion of ECM also promotes the secretion of tumor growth factor beta due to MMP-2, which is involved in tumor proliferation and invasion [56]. The most prominent characteristic of the tumor microenvironment in OSCC was the stimulation of ECM degradation via MMPs activity, which is associated with the release of local growth factors and angiogenesis, further promoting lymph node metastasis [55,56,58].

## 6. MMPs with Identified Roles in the Development and Progression of OSCC

In OSCC, MMPs were identified to have an important role in the early stages of carcinogenesis [55]. Furthermore, they were proposed as therapeutic targets in various malignancies, although their application did not prove to be successful, partially due to their use in patients with already advanced stages of cancer [56]. However, studying the links between the ECM degradation induced by MMPs and various signaling pathways involved in tumor development may reduce the burden of life-threatening metastasis in OSCC [55]. MMPs can be classified according to their substrate and domain structure into collagenases (MMP-1, MMP-8 and MMP-13, mainly digesting collagen types I, II, III, soluble proteins and ECM components), gelatinases (MMP-2 and MMP-9, cleaving collagen types IV, V, XI, laminin), stromelysins (MMP-3, MMP-10 and MMP-11, with similar properties to collagenases, but not degrading interstitial collagen), matrilysins (MMP-7 and MMP-26, interacting with cell surface proteins), membrane-type MMPs (MMP-14, with collagenolytic action) and others (MMP-12) [52,59]. A brief description of the MMPs with established roles in the development of OSCC is presented below.

### 6.1. Collagenases

MMP-1 or collagenase-1 is the earliest identified enzyme from the MMP family, with a role in the remodeling of ECM, being deeply related to inflammation, angiogenesis and metastasis. MMP-1 is secreted in an inactive pro-form and is activated as a result of the propeptide removal by MMP-3 or by a serine protease [60]. Its activity is reduced in healthy

resting tissues and increases in tissue repair and embryonic development, as well as in pathologic states such as chronic cutaneous ulcers and different types of cancer, including oral cancer [27,61]. Based on mass spectrometry evaluation of various salivary proteins, Yu et al. [28] considered that MMP-1 was the most valuable MMP as a potential biomarker for oral cancer detection. Among 28 quantified salivary proteins, these authors [28] reported that MMP-1 had the highest concentration in patients with OSCC along with KNG 1. Furthermore, these two biomarkers also showed high sensitivity (96.7%) and specificity (79.7%), together with a significant power to differentiate OSCC from non-cancerous lesions [28].

MMP-8 is the main collagenase identified in patients with periodontal disease and is responsible for 90–95% of the collagenolytic activity of the gingival crevicular fluid [62]. It was considered an important salivary biomarker in the diagnosis of periodontal disease, although data in the scientific literature regarding its role are still controversial [63,64]. As periodontal disease and OSCC may be found simultaneously in the oral cavity, many studies suggested an association between these conditions, considering that the immunological environment present in periodontal disease could facilitate carcinogenesis [65,66]. Although gingival tissue, the site most affected by periodontal disease, is not a common site for OSCC, this aspect does not eliminate the role of periodontal disease in promoting OSCC development in other areas, such as the floor of the mouth or lateral parts of the tongue, which are more permeable to external carcinogens and more susceptible to malignant transformation [65]. The role of MMP-8 in the progression of OSCC is controversial. It was reported to have protective mechanisms by modifying cell adhesion [67] but also by decreasing tumor cell invasion and migration [68]. Other previous studies showed that high expression of MMP-8 lowered the expression of MMP-1, considered a key collagenase in OSCC development [69,70]. Based on immunohistochemistry, the expression of MMP-8 was mainly present in well-differentiated OSCC, but it showed no correlation with the overall patients' survival [71,72].

MMP-13 is an important enzyme with a role in the digestion of collagen and other components of the ECM [73], thus facilitating tumor invasion. Its expression was significantly higher in oral dysplasia and OSCC tissue compared to normal oral tissue samples, which emphasizes its contribution to the development of OSCC [74]. Moreover, its overexpression has been hypothesized to correlate with poor prognosis in OSCC [74], and its intense activity was proved to favor cancer aggressiveness [75]. It also enhances the metastatic potential of head and neck cancer by increasing angiogenesis [76].

*6.2. Gelatinases*

MMP-2 is a protease with gelatinolytic properties, expressed in renal tubular cells, hepatocytes, and adrenal cortical cells, but also in oral epithelial cells [77]. Its role in the development of the tumor microenvironment is easily understandable since it degrades substrates such as laminin, type IV collagen and proteoglycans, therefore promoting tumor invasion and metastasis. It is mainly secreted by fibroblasts and can be activated by other MMPs, including MMP-1 and MMP-14 [78]. Scientific data reported increased salivary levels of MMP-2 in various conditions of the oral cavity, including dental caries [79]. With regard to OSCC, MMP-2 was detected in the saliva of OSCC patients but not in healthy controls [29]; based on a secretome analysis from OSCC tissue samples, its expression was associated with poor prognosis (low disease-specific survival, disease-free survival and overall survival) [78].

MMP-9 is one of the most complex representatives, which was intensively studied over the years due to its various implications in carcinogenesis [80]. Its main substrates are gelatin, elastin and collagen type IV, the latter being an essential component of the basement membrane, which is usually disrupted in tumor invasion [81]. It was found to be overexpressed in many types of malignancies, including OSCC, as MMP-9-positive cells were especially located in the proximity of disrupted basement membrane sites in tissue samples from PMOLs and OSCC [55]. MMP-9 was shown to be involved in the

progression of dysplasia to cancer, and its polymorphism had a strong correlation with a high risk for OSCC development. Moreover, the constitutive production of MMP-9 by OSCC cells was also identified [82]. Previous data established the role of circular-MMP-9 as a metastasis-promoting gene, involved in OSCC cell migration and development of lymph node metastasis, therefore being associated with worse prognosis [83]. Its important role in OSCC metastasis was also suggested by the increased expression of MMP-9 in metastatic compared to primary OSCC, especially in gingival cancers [84]. Moreover, the increased salivary levels in the early stages of OSCC promoted MMP-9 as a vital highly specific biomarker for early diagnosis and prognosis of OSCC [85].

### 6.3. Stromelysins

MMP-3 was described as a bifunctional protein, which has not only proteolytic properties but is also a transcriptional factor with a major role in tumor progression [86,87]. Its suppression is associated with a significant reduction in tumor growth and cellular migration [87]. The expression of MMP-3 favors tumorigenesis through the following mechanisms: it facilitates cell proliferation, postpones necrosis and preserves the integrity of extracellular vesicles, which are key components in intercellular signaling [88]. Moreover, in OSCC patients, high levels of MMP-3 were correlated with metastasis and poor prognosis [89]. Along with MMP-1, the levels of MMP-3 were significantly modified in the saliva but not in the plasma of OSCC patients, suggesting a high disease-discriminating ability [90]. This is in connection with the local release of these enzymes in the tumor microenvironment; therefore, MMP-3 can be easily detected in saliva but is less represented in systemic circulation [91]. Scientific data suggested that the salivary levels of MMP-3 are also well correlated with the tumor stage [26].

There are limited data on the role of MMP-10 (stromelysin 2) in OSCC. It is usually expressed in epithelial cells and is therefore elevated in tumors of epithelial origin. It is involved in tumor invasion and progression and can activate other MMPs, such as MMP-1, MMP-7, MMP-8, MMP-9 and MMP-13, which have an established role in OSCC [92]. Immunohistochemical studies revealed a higher expression of MMP-10 in head and neck squamous cell carcinomas [93], which is associated with the increased transformation of normal epithelium to OSCC [94] and invasive and lymphatic metastatic potential [95], but also with poor prognosis [93].

### 6.4. Matrylisins

MMP-7 (matrilysin 1) is the smallest enzyme of the MMP family, with multiple interactions with ECM components (type IV collagen, laminin, fibronectin) and cell adhesion molecules (E-cadhrein), but also with inflammatory cytokines (tumor necrosis factor-alpha) [96]. Its overexpression was reported not only in various types of cancer but also in precancerous lesions, which confirmed its role in tumorigenesis [96,97]. It functions like an oncogenic protein by decreasing apoptosis and enhancing angiogenesis, inflammation and cell growth [97]. In OSCC patients, MMP-7 showed elevated levels compared to OLP and healthy individuals; moreover, patients with OLP also had higher levels of MMP-7 than healthy ones, a fact that underlines the role of MMP-7 as an indicator of early malignant transformation [98]. In addition to this, in tongue squamous cell carcinoma, the high expression of MMP-7 accelerated in vitro tumor cell growth and migration and increased the risk of nodal metastasis in vivo, while its suppression produced the opposite effect [96].

### 6.5. Others

MMP-12 might be expressed by macrophages and tumor cells of epithelial origin, which influences its effect on tumor progression. When present in squamous tumor cells, it favors tumor aggressiveness, while when originating from macrophages, it displays inhibitory properties on tumor angiogenesis and growth [99]. When measured from serum along with MMP-1, MMP-8, MMP-10 and MMP-13, MMP-12 exhibited the highest con-

centrations in OSCC patients, with 80% sensitivity and 78.9% specificity, which enabled it as a possible diagnostic marker [100]. High levels of MMP-12 were found in the saliva of patients with various forms of cancer, which made it a possible candidate for tumor screening [101]. A summary of the interactions between different MMPs and ECM components in OSCC is illustrated in Figure 1.

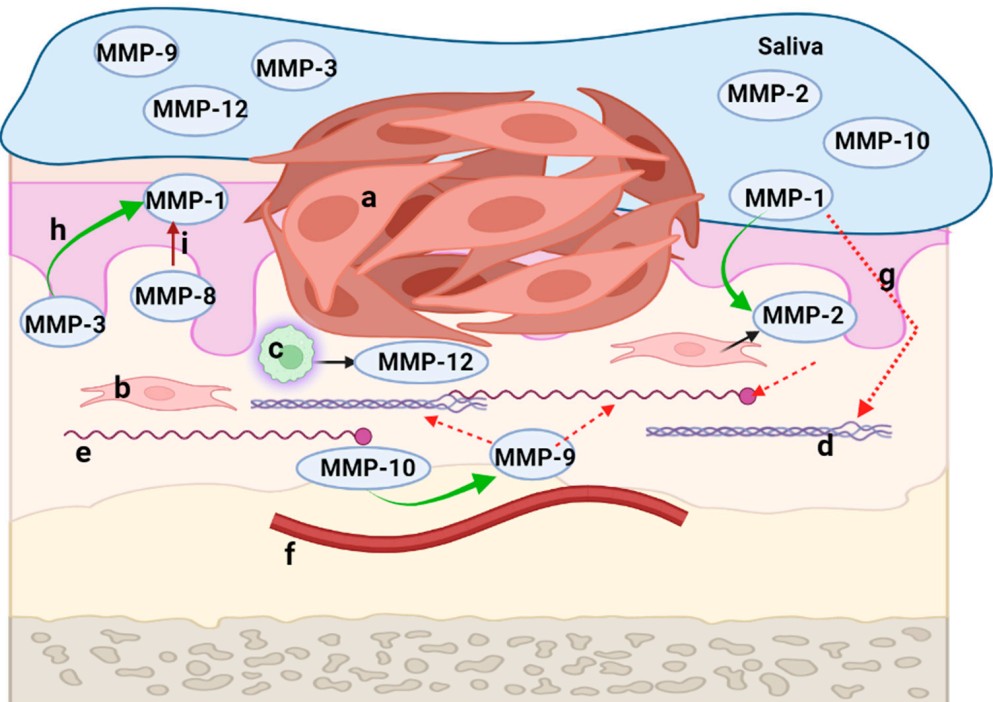

**Figure 1.** MMPs activity in the tumor microenvironment of OSCC facilitates invasion through the digestion of ECM components. MMP-1 degrades collagen type I and activates MMP-2; MMP-2 is released from fibroblasts at the invasive front and degrades collagen type IV. MMP-3 contributes to tumor invasion by activating MMP-1. MMP-8 (by inactivating MMP-1) and MMP-12 (when released from macrophages) may have protective effects. MMP-9, which is activated by MMP-10, degrades both type I and type IV collagen. All these MMPs can also be found in the saliva of patients with OSCC. (Figure legend: a = primary tumor, b = fibroblast, c = macrophage, d = collagen type I, e = collagen type IV, f = blood vessel, g = interrupted red arrow, suggesting degradation effect, h = green arrow, suggesting activation, i = continuous red arrow, suggesting inhibition).

## 7. The Use of Salivary MMPs as Biomarkers for OSCC

The improved diagnostic technologies based on molecular biology highlighted the potential use of salivary biomarkers in the early detection of OSCC [102]. Proteomic platforms described around 200 peptides and proteins present in saliva and differentiated them into physiological and pathological conditions; furthermore, the salivary proteome for oral lesions and OSCC has been characterized [103], and over 100 potential salivary biomarkers for OSCC were identified [104]. Based on their central role in tumor progression, several studies focused on the salivary detection of MMPs [26,27,29–36,105]. These reported significantly higher concentrations of mainly MMP-1, MMP-2, MMP-3 and MMP-9, but also of other MMPs in the saliva of patients with OSCC, compared to healthy controls, which were also correlated with the tumor stage [26,105]. A summary of findings related to the salivary levels of MMPs detected in patients with PMOLs, and OSCC is presented in Table 2.

**Table 2.** Significance of the salivary levels of MMPs in PMOLs and OSCC.

| MMPs | | Study Findings and Clinical Relevance |
|---|---|---|
| Collagenases | MMP-1 | Higher salivary levels in OSCC patients compared to healthy controls [26,29]; Higher levels in the saliva of patients with OSCC compared to that of patients with OBM and CPD [29]; Significant ability to discriminate OSCC from non-tumor lesions [27]; Correlation of the salivary levels with OSCC tumor stage and nodal metastasis [26,27]; Higher salivary levels in HPV-negative OSCC and higher differences between cases and controls in OSCC compared to oropharyngeal tumors [26]. |
| | MMP-8 | Identified upregulation in the saliva of patients with OSCC [30]. |
| | MMP-13 | Identified upregulation in the saliva of patients with OSCC [30]; Lack of difference between the salivary levels detected in patients with OSCC and OLP [31]. |
| Gelatinases | MMP-2 | Higher levels in the saliva of patients with OSCC compared to that of patients with OBM, CPD and healthy controls [29]. |
| | MMP-9 | Higher salivary levels in patients with OSCC compared to patients with PMOLs and healthy controls; higher salivary levels in patients with PMOLs compared to healthy controls [32,33]; Greater sensitivity and specificity compared to serum [32]; Salivary levels significantly decrease (up to 80%) after surgical excision of OSCC [34]; Salivary levels increased in poorly differentiated OSCC in comparison with moderate and well-differentiated tumors [35]. |
| Stromelysins | MMP-3 | Identified upregulation in the saliva of patients with OSCC [30]; Increasing salivary levels from the reticular to the erosive form of OLP and then to early and advanced stages of OSCC [31]; Higher salivary levels in patients with OSCC compared to healthy controls and correlation with disease stage [26]; Higher salivary levels in HPV-negative OSCC and higher differences between cases and controls in OSCC compared to oropharyngeal tumors [26]. |
| | MMP-10 | Higher levels in the saliva of patients with OSCC compared to that of patients with OBM, CPD and healthy controls [29]; Identified upregulation in the saliva of patients with OSCC [30]. |
| Others | MMP-12 | Higher levels in the saliva of patients with OSCC compared to that of patients with OBM, CPD and healthy controls [29]; Higher salivary levels among patients with OSCC, patients with OSF and healthy controls [36]. |

    Most of the studies evaluating the role of MMPs in pathological conditions focused on malignancies; however, more recent data emphasized their involvement in various diseases, which all have in common processes of inflammation, tissue remodeling, angiogenesis, cell growth and differentiation. The interrelation of MMPs with the tumor microenvironment is complex, as some of them (MMP-1, MMP-9) are also induced by external carcinogens such as tobacco or betel quid chewing [106]. Although their role in OSCC development and progression is relatively well established, more research on the possible routine application of salivary MMPs in the detection and monitoring is warranted, as saliva proved excellent qualities as a diagnostic fluid. This is not only due to its non-invasive character and proximity to the tumor site but also due to its lower concentration of inhibitory substances and safer handling compared to plasma [38,107]. Despite their identified contribution in differentiating OSCC from healthy controls (MMP-1, MMP-3, MMP-9) and their increasing levels in association with tumor stage by salivary assessments, more research is also needed

in order to validate associations of the salivary MMPs with histological grading and lymph node metastasis [108].

## 8. Difficulties Associated with the Early Diagnosis of OSCC

The detection of OSCC in the early stages is difficult, as most of the cases are asymptomatic and, by clinical examination, look similar to PMOLs such as leukoplakia, erythroplakia or OLP [109,110]. Moreover, the distinction between different types of PMOLs is difficult, and from a clinical point of view, the majority of cases will have an imprecise diagnosis [111]. Leukoplakia is a frequently encountered lesion with a malignant transformation rate of 0.13–34.0% [112,113]. Conversely, erythroplakia has a lower prevalence but significantly higher rates of malignant transformation, ranging between 14 and 50% [114]. Moreover, previous data showed that almost 90% of patients with erythroplakia exhibited foci of OSCC from the first biopsy [115,116]. OLP could also increase the risk of oral cancer [117], with a malignant transformation rate of about 1.4% [118,119]. However, OSCC can also have a "de novo" appearance without any previous alteration of the oral epithelium [120]; the detection delay has a negative impact on the survival rate and allows this very aggressive malignancy to produce metastases in distant organs, identified in approximately 50% of the patients at the moment of diagnosis [121,122]. Therefore, early detection of OSCC and the screening of all types of PMOLs become crucial in prevention and treatment outcomes [123]. In order to overcome the problem of early diagnosis, various diagnostic methods for screening have been proposed, such as exfoliative cytology (cytobrush) [123–125], the use of vital dyes [46], evaluation of oral mucosa alteration using light emission sources (chemiluminescence and autofluorescence) [126,127] and more recently, molecular biomarkers and salivary assays [128,129], which seemed promising methods, with high potential for becoming diagnostic tests [37].

## 9. Strengths and Limitations

The present review has an important clinical value, as it shows that specialists can obtain rapid and useful information regarding the salivary levels of MMPs by using a simple, cost-effective and well-tolerated sampling method. The studies included in our analysis used accessible laboratory tests, such as ELISA, which is more convenient in comparison with expensive genetic technologies. Although the gold standard for PMOLs and OSCC diagnosis remains the biopsy with the histopathological examination, salivary levels of MMPs may represent an adjuvant method for early diagnosis but also for patients' monitoring and prognosis after treatment. However, the strength of evidence of our paper is limited due to the narrative character of the review, based on 11 articles that included mostly small samples of patients. Moreover, systematic reviews with meta-analysis are needed in order to reduce bias caused by the subjectivity and heterogeneity of the selected studies.

## 10. Final Considerations

Various studies have evaluated the potential role of different MMPs detected in saliva in the diagnosis and prognosis of OSCC. MMP-1 and MMP-3 had significantly higher salivary levels in OSCC, correlated with tumor stage, compared to the nearly undetectable levels in the saliva of control subjects. Moreover, more extensive research of MMP-1 in larger cohorts established it as a valuable marker for screening high-risk patients, diagnosis and monitoring of OSCC. In the same manner, MMP-9 was also able to distinguish the early stages of OSCC, with a higher sensitivity when measured in saliva compared to serum. The high power of discrimination between OSCC and healthy subjects, along with the correlation with tumor stage, are important features that characterize MMPs as promising adjuvant tools in the diagnosis and monitoring of OSCC. However, their application in clinical practice is still restricted due to the following limitations: the small study groups of patients included in the majority of studies, the deficient adjustment of all covariates, certain bias factors such as concomitant periodontal disease, different protocols of saliva sampling and multiple assay kits. Therefore, larger studies aiming to correlate the salivary

levels of MMPs with gold standards of OSCC management (white-light evaluation for screening, radiologic investigations for staging and monitoring) are needed in order to validate the routine use of these biomarkers in the clinical setting.

**Author Contributions:** Conceptualization, M.M. and A.M.P.; validation, M.M.; investigation, A.M.P.; data curation, M.M. and A.M.P.; writing—original draft preparation, M.M. and A.M.P.; writing—review and editing, M.M. and A.M.P.; visualization, M.M.; supervision, M.M. All authors have read and agreed to the published version of the manuscript.

**Funding:** This research received no external funding.

**Institutional Review Board Statement:** Not applicable.

**Informed Consent Statement:** Not applicable.

**Data Availability Statement:** Not applicable.

**Conflicts of Interest:** The authors declare no conflict of interest.

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
