# Peer review of "The Use of Salivary Levels of Matrix Metalloproteinases as an Adjuvant Method in the Early Diagnosis of Oral Squamous Cell Carcinoma: A Narrative Literature Review"

_cimb, doi:10.3390/cimb44120430_

Round 1

Reviewer 1 Report

The manuscript intitled “The use of salivary levels of matrix metalloproteinases as an adjuvant method in the early diagnosis of oral squamous cell carcinoma” discusses a relevant topic. The text is clear and well-written. However, the authors should consider some important points to improve the manuscript.

1.       Title: We suggest to add at the end of the title “A Narrative Literature Review” OR “State of Art”;

2.       Abstract: It is important to mention the limitations of the study at the end of the abstract;

3.       Introduction: Good.

4.       Material and Methods:  This section must be briefly described. The authors have to mention the criteria for selecting the manuscripts (relevance, date of publication), the period of the search and the strategy of matching the descriptors used in the search too.  It would be good state what type of study they selected: case reports, randomized clinical trials, prospective studies, cross-sectional studies?

5.       The manuscript discussed some important topics and its publication will surely contribute for the understanding of how the use of salivary levels of matrix metalloproteinases will help early diagnosis of oral squamous cell carcinoma.

6.       Please, replace “Conclusions” to “Final Considerations”. As it is a narrative review the strength of evidence is weak. The authors should also recommend systematic reviews with meta analysis.

Author Response

Dear Reviewer,

Thank you for evaluating our paper, in the attachment you may find our response to your comments.

Reviewer 2 Report

This review aimed to provide up-to-date information regarding the role of salivary levels 73 of MMPs as complementary tools for the early diagnosis of PMOLs and OSCC, based on 74 data available on these molecules’ contribution to OSCC progression and on the use of 75 saliva as a diagnostic fluid. For that purpose, a narrative literature review was conducted.

The manuscript is well-written and provides enough information related to the review topic and could be interesting for the journal readers. However, I have some minor comments.

1.     Keywords: They should be MeSH terms

2.     Please mention in the methods section (data collection) what kind of findings of what items will be described/explored in the narrative review.

3.     Please mention the limitations of this narrative review (considering the importance of carrying out systematic reviews by eliminating several aspects related to the subjectivity and the heterogeneity of the selected studies).

4.     Please mention clinical and practice recommendations (it could be necessary for the readers)

Author Response

(The authors gave the same response as above.)

Reviewer 3 Report

I thank the editors for allowing me to review this manuscript.

The manuscript is interesting, it clearly and completely describes the role, known up to now, of MMPs in the OSCC.

However there are some points to clarify:

In the introduction the statement ", the incidence of OSCC is expected to rise in the next 20 years with up to 40%, along with a correlated rise in mortality" would be useful to explain the reason for this expected increase.

In data collection:

How many studies have been evaluated in total?

What were the exclusion criteria?

In table 1, bibliography 30:

In Conclusions, particularly "MMP-9 can be useful biomarkers for the diagnosis and prognosis of OSCC," but in the Results this concept is not clear.

Sincerely

Author Response

(The authors gave the same response as above.)
